# β-Lactam TRPM8 Antagonists Derived from Phe-Phenylalaninol Conjugates: Structure–Activity Relationships and Antiallodynic Activity

**DOI:** 10.3390/ijms241914894

**Published:** 2023-10-04

**Authors:** Cristina Martín-Escura, M. Ángeles Bonache, Jessy A. Medina, Alicia Medina-Peris, Jorge De Andrés-López, Sara González-Rodríguez, Sara Kerselaers, Gregorio Fernández-Ballester, Thomas Voets, Antonio Ferrer-Montiel, Asia Fernández-Carvajal, Rosario González-Muñiz

**Affiliations:** 1Instituto de Química Médica (IQM-CSIC), 28006 Madrid, Spain; 2Alodia Farmacéutica SL, 28108 Alcobendas, Spain; 3IDiBE, Universidad Miguel Hernández, 03202 Elche, Spain; 4Facultad de Medicina, Instituto Universitario de Oncología del Principado de Asturias (IUOPA), Universidad de Oviedo, Julián Clavería 6, 33006 Oviedo, Spain; 5Laboratory of Ion Channel Research, Department of Cellular and Molecular Medicine, VIB Center for Brain and Disease Research, KU Leuven, Herestraat 49 Box 802, 3000 Leuven, Belgium

**Keywords:** TRPM8 channel, β-lactams, antagonists, docking, antinociceptive activity

## Abstract

The protein transient receptor potential melastatin type 8 (TRPM8), a non-selective, calcium (Ca^2+^)-permeable ion channel is implicated in several pathological conditions, including neuropathic pain states. In our previous research endeavors, we have identified β-lactam derivatives with high hydrophobic character that exhibit potent and selective TRPM8 antagonist activity. This work describes the synthesis of novel derivatives featuring *C*-terminal amides and diversely substituted *N*′-terminal monobenzyl groups in an attempt to increase the total polar surface area (TPSA) in this family of compounds. The primary goal was to assess the influence of these substituents on the inhibition of menthol-induced cellular Ca^2+^ entry, thereby establishing critical structure–activity relationships. While the substitution of the *tert*-butyl ester by isobutyl amide moieties improved the antagonist activity, none of the *N*′-monobencyl derivatives, regardless of the substituent on the phenyl ring, achieved the activity of the model dibenzyl compound. The antagonist potency of the most effective compounds was subsequently verified using Patch-Clamp electrophysiology experiments. Furthermore, we evaluated the selectivity of one of these compounds against other members of the transient receptor potential (TRP) ion channel family and some receptors connected to peripheral pain pathways. This compound demonstrated specificity for TRPM8 channels. To better comprehend the potential mode of interaction, we conducted docking experiments to uncover plausible binding sites on the functionally active tetrameric protein. While the four main populated poses are located by the pore zone, a similar location to that described for the N-(3-aminopropyl)-2-[(3-methylphenyl)methoxy]-N-(2-thienylmethyl)-benzamide (AMTB) antagonist cannot be discarded. Finally, in vivo experiments, involving a couple of selected compounds, revealed significant antinociceptive activity within a mice model of cold allodynia induced by oxaliplatin (OXA).

## 1. Introduction

Somatosensory neurons are capable of perceiving changes in external temperature through thermosensitive TRP ion channels [1,2]. These thermoTRP channels play pivotal roles in human physiology [3,4], but they are also implicated in human pathology [5,6], making them important targets for therapeutic interventions [7,8]. Among the thermoTRP family, TRPM8 stands out as a notable member. Initially identified as a marker for prostate cancer [9], TRPM8 was subsequently recognized as a sensory ion channel, which is crucial for cold sensing and pain transmission [10,11,12,13,14]. This cationic channel, with a particular preference for Ca^2+^ permeability, operates through multimodal activation mechanisms encompassing cold temperatures (> 15 < 28 °C), voltage changes, osmolality alterations, and cooling compounds of natural origin, such as menthol and icilin [15,16].

Numerous experimental findings have correlated the expression, function and mutation of TRPM8 and various pathological conditions [17,18]. Thus, this channel is involved in pain perception and modulation, and TRPM8 inhibition and channel desensitization (by agonists pressure) have been shown to alleviate both acute and chronic pain conditions [12]. For instance, painful bladder hypersensitivity and peripheral neuropathy resulting from chemotherapy have been linked to elevated TRPM8 expression and/or function [18,19,20]. Notably, studies using TRPM8 agonists suggest their potential for treating patients with neuropathic ocular pain, while TRPM8 blockade has proven effective as well in alleviating pain associated with severe dry eye disease [21]. In pulmonary diseases, TRPM8 channels have been related to asthma and chronic obstructive disease (COPD). Accordingly, the downregulation of TRPM8 expression has been connected with enhanced vasoreactivity in pulmonary hypertension in mice [22]. Additionally, certain *TRPM8* and *TRPA1* polymorphisms have been implicated in the pathogenesis of COPD [23]. Moreover, it has been observed that exposure to cold air triggers airway inflammatory responses and alters TRPM8 expression. Interestingly, the knockdown of TRPM8 has been shown to mitigate respiratory hypersensitivity in asthmatic individuals [24]. 

The altered expression of TRPM8 channels also underlies the mechanisms driving tumor progression in various types of cancers, including prostate, pancreas, breast, colon, and lung. However, the impact on expression varies based on the tumor type and stage of advancement, with contrasting patterns of overexpression or downregulation, depending on the specific context [25]. 

Since its discovery in 2001, a wide array of chemically diverse compounds have been identified as modulators (agonists and antagonists) of the TRPM8 channel [7,8,26] with the most important contribution from several pharmaceutical companies. Notably, among these efforts, compounds like PF-05105679 and AMG333, both orally active TRPM8 antagonists, advanced into clinical trials with the aim of exploring their potential for treating cold pain hypersensitivity and relieving migraine symptoms, respectively [27,28]. However, these promising compounds failed to progress beyond phase I trials [29]. Volunteers participating in the trials reported significant side effects, including intense sensations of heat in the hands, arms, and mouth [30], which could likely be attributed to insufficient selectivity against heat-activated TRP channels. Other notable drawbacks of these clinical candidates included dysesthesia, paresthesia, and dysgeusia. These limitations might be connected to the pivotal role of the TRPM8 channel in regulating core body temperature as well as other less understood physiological functions [31,32]. This context could potentially restrict the use of TRPM8 modulators as oral or systemic medications. Nonetheless, the prospect of utilizing these modulators as locally administered or topical treatments for TRPM8-related conditions remains largely underexplored.

In previous research, our team has identified several β-lactam derivatives exhibiting potent and selective TRPM8 antagonist activity [33,34]. Furthermore, some of these derivatives have demonstrated antinociceptive properties in various animal models [35]. In the present study, our focus lies on deepening our understanding of the structure–activity relationships within this compound family. Specifically, we investigate the introduction of different amides at the C-terminus and explore the potential of various substitutions in N-monobenzyl derivatives (Figure 1). We were particularly interested in enhancing the TPSA of the previously identified *N*,*N*-dibenzyl derivative **1**, to reduced absorption capabilities, and thereby to open up possibilities for investigating local or topical applications.

## 2. Results and Discussion

### 2.1. 1,4,4-Trisubstituted β-Lactam Derivatives

For the first exploratory structure–activity relationships, we decided to prepare and evaluate diastereoisomeric 1,4,4-trisubstituted β-lactam derivatives, which can be more easily prepared than enantiomerically pure analogues. In previous works, we have demonstrated that both 4-*S*- and 4-*R*-β-lactams rendered TRPM8 antagonists [36]; therefore, we expected that 4-*R*,*S*-diastereoisomeric mixtures could be active and allow us to discern among important substituents. Firstly, we proposed the incorporation of different amides in the 4-carboxylate group, including structural diversity with an alkyl group, a benzyl group, a pyridine group and other related heterocyclic systems, to slightly increase the TPSA values (Appendix A). The six amines selected for the formation of the corresponding amides were *iso*-butylamine, benzylamine, 3,4-dichlorobenzylamine, 2- and 3-picolines (2-, 3-Pic) and 3-aminopyridine (3-Py). In a previous publication on Asp-derived β-lactams, amide derivatives from 4-pyridine and 4-picoline showed lower activity than the corresponding 3-aminopyridyl analogues [33]. Therefore, the 4-picolyl derivative was discarded in this case, while the 2-picoline derivative was prepared, since it had not been explored previously. In addition, pyridyl- and picolyl-derived amides incorporate two protonable nitrogens, which could increase the aqueous solubility of final molecules. The 3,4-dichlorobenzylamine was selected because this substitution is present in a kappa opioid agonist with antagonist-like properties in a mouse model related to TRPM8 activation by icilin [37]. The saponification of β-lactam **2a**,**b** (see Appendix A for synthetic details) afforded the carboxylic acid intermediate **3ab** (**a**:**b** = 1:1.8), which was then coupled with the selected amines, in the presence of Bromo-tris-pyrrolidino-phosphonium hexafluorophosphate (PyBroP), to allow the preparation of amides **4a**,**b**–**9a**,**b** (Figure 1). These amides were isolated in different diastereoisomeric ratios (Appendix A), which was most probably due to different conversions of the diasteromeric acids **3a** and **3b** or to the enrichment in a particular diastereoisomer during the purification process. 

Another way to increase TPSA values is the removal of one of the *N*-benzyl groups, while the incorporation of different substitutions on the remaining *N*-benzyl moiety could serve to fine-tune the antagonist activity (Appendix A). To this end, the hydrogenolysis of β-lactam **2a**,**b**, using Pd(OH)_2_/C as a catalyst in the presence of HCl, gave rise to the primary amine derivative **10a**,**b** in its hydrochloride form (1:2.5, **a**:**b** diastereoisomer ratio). The slight enrichment in the major diastereoisomer (**b**) compared to the starting material is due to the formation of the (3S,6S)-3,6-dibenzyl-1,4-diazabicyclo[4.2.0]octane-5,8-dione **11a** (configuration determined through nuclear Overhauser effect, NOE, experiments) from minor isomer **10a** (Figure 2). This secondary product was also formed during the reductive amination reaction of compound **10a**,**b** with aldehydes to the corresponding monobenzyl derivatives, complicating the purification steps. To avoid this issue, most reactions in this section were performed with the *iso*-butylamide derivative **12a**,**b**, which was synthesized from **4a**,**b**. Differently substituted benzaldehydes, containing alkyl or benzyl groups, and electron attracting or withdrawing groups alongside 2-naphtaldehyde were used for preparation of the N-monobezyl derivatives (Figure 2, Appendix A). Following some of the activity initial results detailed later on, most substituents on the benzyl group are at position 3 of the phenyl ring, but some 4-substituted analogues were also prepared to verify preliminary outcomes. Thus, amine derivative **10a**,**b** was treated with the corresponding benzaldehyde to form the corresponding intermediate imines, which were subsequently reduced with NaBH_3_CN to afford β-lactam *N*-monobenzylated derivatives **13a**,**b**–**17a**,**b** (Figure 2, Appendix A). Similarly, the reaction of **12a**,**b** with different aldehydes, containing an aliphatic (Me), two aromatic (Ph, Nf), all halides (F, Cl, Br, I), donor groups (OMe, OPh, OBn) and electron acceptors (CN, NO_2_) substituents afforded the corresponding monobenzyl derivatives **18a**,**b**–**31a**,**b** (Figure 2, Appendix A).

The diastereomeric 1,4,4-trisubstitutes β-lactams were evaluated as modulators of the TRPM8 channel by calcium microfluorimetry assays. These assays use a Human Embryonic Kidney 293 (HEK293) cell line that stably overexpressed rat TRPM8 channel (*r*TRPM8). Menthol was used as the agonist during these experiments, whereas AMTB and model compound **1** served as prototype antagonists for comparison. Three separated experiments in triplicate were used. The maximum inhibitory rate on calcium flow reached 100% at 50 μM concentration for most of the evaluated compounds.

Within these assays, β-lactam **2a**,**b**, chosen as a pivotal intermediate, displayed submicromolar potency as an antagonist of the TRPM8 channel (Table 1). Among the amide series, the most favorable IC_50_ values were achieved with compounds **4a**,**b** and **5a**,**b**, suggesting that the methyl ester of **2a**,**b** can be substituted by an aliphatic (**4a**,**b**) or benzylic (**5a**,**b**) amide with only marginal loss of activity (Table 1). In contrast, the introduction of two chlorine atoms into the benzylamide group (**6a**,**b**) causes a notable reduction in antagonist activity. The amide derived from 2-picoline (**8a**,**b**) exhibited micromolar potency, while the 3-aminopyridyl (**7a**,**b**) and 3-picolyl (**9a**,**b**) derivatives demonstrated lesser potency. When comparing these results to the model antagonist AMTB, it becomes apparent that the 4-CO_2_Me derivative **2a**,**b** and its amide counterparts **4a**,**b** and **5a**,**b** exhibit antagonist activity one order of magnitude more potent.

Within the monobenzyl series, all compounds exhibit IC_50_ values in the micromolar range, displaying slight variation across most of them (1.10 to 12.1 μM) and lower potency in comparison to the dibenzyl model compounds **2a**,**b** and **4a**,**b** (Table 2). The importance of having at least one benzyl-type group at the N-terminal amino moiety is demonstrated by the substantial decline in antagonist activity observed in **11a** (with less than 50% inhibition at 50 μM), although the conformational restriction could also contribute to the lower activity.

Regarding the substituents on the benzyl group, placement at the *meta* (3-) position seems slightly preferred over *para* (4-) positioning. This observation is drawn from the comparison between compounds **14a**,**b** vs. **15a**,**b**, as well as between **20a**,**b** and **21a**,**b** vs. **30a**,**b** and **31a**,**b**, respectively (Table 2). Large substituents around position 3, such as Ph (**20a**,**b**), Cl (**22a**,**b**), Br (**16a**,**b** and **23a**,**b**), I (**24a**,**b**), and the condensed 2-naphthyl moiety (**17a**,**b** and **32a**,**b**), appear to enhance activity. Generally, strong electron-donating groups like OMe (**25a**,**b**) and OBn (**27a**,**b**), which result in an increased TPSA up to 70 A^2^ (Appendix A), exhibit slightly lower potency than their corresponding unsubstituted analogue (**13a**,**b**). On the contrary, a distinct behavior is noted between the two analogues bearing strong electron-withdrawing groups (CN and NO_2_). While the nitrile-bearing derivative **28a**,**b** ranks as the less effective compound in this series (IC_50_ = 12.1 µM), the 3-NO_2_ analog **29a**,**b** maintains the one-digit micromolar antagonist activity akin to the unsubstituted counterpart. The calculated TPSA values of these latter compounds have increased by more than 25 and 55 units, respectively, when compared to the unsubstituted analogue **13a**,**b** (as shown in Appendix A). However, compound **29a**,**b** was not pursued further due to the potential interference posed by the NO_2_ group, which is signaled as a Pan-Assay Interference Compound (PAIN) alert [38].

All these observations on structure–activity relationships must be approached with the required caution, considering the variations in the ratio of diastereomers among different compounds (Appendix A). In general, the *N*-monobenzylated derivatives exhibit a reduced activity ranging between one and two orders of magnitude when compared to the corresponding *N*,*N*-dibenzylated model compounds. The introduction of different substituents on the phenyl ring does not result in significant impacts on activity, independently of the substituent size and electronic character. Even large substituents (such as Ph or Br) on the *N*-monobenzyl moiety failed to compensate for the absence of the second benzyl group.

### 2.2. 1,3,4,4-Tetrasubstituted, Enantiopure β-Lactam Derivatives

The challenges in making precise comparisons among compounds of the previous 1,4,4-trisubstituted series, due to the variability in the diastereoisomeric ratio and the requirement to study pure isomers for further pharmaceutical development, prompted us to transition to a new series of diastereo- and enantiopure β-lactam derivatives. In this newly designed series, the most promising substituents identified earlier were incorporated onto a 1,3,4,4-tetrasubstituted β-lactam scaffold. These enantiopure β-lactams can be synthesized using chiral 2-chloropropanoic acid derivatives as previously reported [35,39].

Staring from compound **1**, the tert-butyl ester underwent acidic hydrolysis to yield the free carboxylic acid intermediate **33**. This intermediate was subsequently coupled to isobutyl, benzyl and 2-picolyl amines to generate the corresponding amides, **34**–**36**, with very good yield (Figure 3). The hydrogenolysis of compound **1**, using Pd(OH)_2_ as a catalyst, resulted in a mixture of the monobenzyl derivative **37** and the fully deprotected amino analogue **38**, which were separated using column chromatography. A similar reaction, starting from compound **34**, rendered the free amino derivative **39** in almost quantitative yield. Further treatments of compounds **38** and **39** with differently substituted benzaldehydes, followed by a reduction of the resulting imine intermediates with NaBH_3_CN, led to the synthesis of the expected *N*-monobenzyl derivatives **40**–**45** (Figure 3, Appendix A).

The initial biological evaluation of this series of 1,3,4,4-tetrasubstituted β-lactams was performed by calcium microfluorimetry assays using the indicated HEK293 cell line overexpressing *r*TRPM8 channels. As outlined in Table 3, submicromolar IC_50_ values were achieved for the three prepared amide derivatives with their antagonist activities ranking as follows: **34** (R^1^ = *^i^*Bu) > **35** (R^1^ = Bn) > **36** (R^1^ = CH_2_(2-Pic)). These levels of potency surpass that of the O*^t^*Bu model compound **1** and are one order of magnitude higher than the reference antagonist AMTB. In line with previous findings, most of the *N*-monobenzylic compounds displayed reduced potency compared to their dibenzyl analogues. Furthermore, substitution at the 3-position of the benzyl aromatic ring appears to be slightly favored over the 4-position (comparison of **40** vs. **41**). Compounds with 3-Ph (**42**), 3-Br (**44**) and 2-Nf (**45**) exhibited similar micromolar activities. Unfortunately, our efforts to increase the TPSA values by the incorporation of additional heteroatoms in amides and *N*-monobenzyl derivatives resulted mostly in lower potencies compared to model compound **1**. Interestingly, amides **34** and **35** improved the TRPM8 antagonist activity, although the increases in TPSA values are tiny for these compounds (Appendix A).

### 2.3. Activity in hTRPM8 Channels and Electrophysiology Assays

To further characterize this family of compounds, calcium microfluorimetry assays were conducted using the HEK293 cell line, which expresses the human isoform of the TRPM8 channel. Amide derivatives **34** and **35** were selected, and their outcomes were compared with those of the model compound **1** (Table 4). These new β-lactams exhibit a somewhat diminished antagonist activity, as evidenced by higher IC_50_ values, when evaluated in *h*TRPM8 as compared to *r*TRPM8. Since there is a high degree of homology between both orthologues (see Appendix A), especially in the ligand-binding site, as well as in the subsites suggested by our docking studies, this small deviation in the antagonist activity could be attributed to some dissimilarity in the regulation pathways. This behavior is not particular of these β-lactams, but it is similar to those observed for other TRPM8 antagonists [40]. 

To confirm the antagonistic activity of the most potent compounds, electrophysiology studies were conducted using the Patch-Clamp technique on single HEK293 cells expressing the rat TRPM8 channel (Table 4, Figure 2). A recent research study using Patch-Clamp, among other techniques, established that menthol occupies a cavity within the TRPM8 S1–S4 transmembrane helix, with the OH group forming an H-bond with the δ-NH of R842 residue, and Van der Waals interactions among the methyl groups of the *^i^*Pr moiety and residues I846 and L843 [41]. Tyr 745, Y1004 and R1007 residues have also been revealed to be important for TRPM8 activation by cooling compounds [42].

As depicted in Figure 2A, perfusion with 100 µM menthol (shown in blue) resulted in a strongly outward rectifying ion current. This is evident by a negligible current at negative potentials and a linear intensification of the ohmic current at positive voltages. When applying compound **34** at a concentration of 5 µM, a noticeable reduction in menthol-induced activity at depolarizing voltages was observed (indicated by the red line) compared to menthol alone. Dose–response relationships for both tested compounds were established through triplicate experiments at various concentrations, applying a holding voltage of −60 mV (as shown in Figure 2B). These compounds effectively blocked the TRPM8-mediated and menthol-induced responses. In these assays, both derivatives displayed higher potency than the model β-lactam **1**, being the *iso*-propylamide derivative **34** the most potent, with efficacy in the low nanomolar range (Table 4).

We also investigated the effect of β-lactams derivatives on the resting membrane potential (RMP) of dorsal root ganglia (DRG) sensory neurons. Under current clamp conditions in Patch-Clamp assays, small sensory neurons displayed an RMP of ~−50 mV (Figure 3A, upper left trace). Exposure of the neurons to 10 µM of compound **34** or **35** did not alter the RMP of these neurons (Figure 3A, lower right traces and Figure 3B). As a control to ensure that neurons were able to exhibit action potential firing, they were exposed to 100 µM menthol (Figure 3A, right trace, and Figure 3B).

### 2.4. Activity in Other TRP Channels and Pain-Related Peripheral Receptors

The effects of compound **34** have been evaluated on different ion channels and receptors implicated in pain-related processes to assess the selectivity for TRPM8 channels. First, some TRP channels that play a role in temperature integration and nociception were selected [29,43], including *h*TRPV1, *h*TRPV3, *h*TRPA1 and TRPM3. Additionally, the evaluation encompassed the *h*ASIC3 channel with widespread distribution within the peripheral nervous system and contributions to the excitability of primary sensory neurons [44]. In each instance, the compound was subjected to evaluation at a consistent concentration of 10 µM. As outlined in Table 5, it is evident that compound **34** did not exhibit noteworthy modulatory effects on any of the assessed TRP and ASIC3 channels. Importantly, amide derivative **34** did not show activation on the TRPV1 channel, which was about 30% in the model compound **1** [35]. Compound **35** underwent evaluation as a potential antagonist for the TRPM3 channel, revealing a non-significant degree of inhibition (4.1 ± 10.8) in the Ca^2+^ influx induced by pregnenolone sulfate.

Furthermore, the investigation extended to measuring the binding affinity of compound **34** to calcitonin gene-related peptide *h*CGRPR, cannabinoid *h*CB2 and muscarinic *h*M3 receptors, which are all relevant to different pain conditions [45,46,47,48,49]. As shown in Table 5, this compound shows inability to significantly displace the specific radioligands for *h*CB2, *h*M3, and *h*CGRPR receptors.

### 2.5. Antinociceptive Activity in a Mouse Model of Chemotherapy-Induced Cold Allodynia

It is established that treatments with the chemotherapeutic drug oxaliplatin (OXA) induce painful peripheral neuropathy, known as CIPN, which becomes more pronounced in cold conditions [50]. The TRPM8 channel has been implicated in mouse models of CIPN pain induced by OXA [19,51]. In fact, the OXA-induced peripheral neuropathy mouse model is being used for the pharmacological characterization of TRPM8 antagonists. For this peripheral neuropathy assay, mice were subcutaneously (s.c.) injected with OXA at a dose of 6 mg/kg dose on days 1, 3 and 5 to male mice to induce increased sensitivity to cold, which can be monitored through the acetone drop test.

Consequently, compounds **34** and **35** have been tested for their ability to alleviate OXA-induced cold hypersensitivity after intraplantar administration (i.pl., 3, 1 and 0.1 μg). As depicted in Figure 4A,B, both compounds **34** and **35** exhibited a dose-dependent and significant reduction the cold-induced paw licking. This effect was observed 15 min post-administration and sustained up to 60 min. These findings parallel those reported for a biphenylamide *N*-spiro[4.5]decan-8-yl derivative, which demonstrated antiallodynic activity lasting up to 60 min at a 1 µg dose, despite possessing higher nanomolar potency on rat TRPM8 [52]. Notably, compounds **34** and **35** displayed greater potency and a longer lasting action compared to another group of TRPM8 antagonists with an imidazo[10,50:1,6]pyrido[3,4-b]indole-1,3(2H)-dione central scaffold [53]. The superior in vivo activity of this β-lactam family might be attributed to their distinct mode of interaction as suggested by modeling studies.

### 2.6. Insights into the Mode of Interaction of Selected β-Lactams with the TRPM8 Channel

To shed light on the mechanism by which β-lactam amides and monobenzyls interact with the TRPM8 channel, we select compounds **34**, **35**, and **37**, which represent the most potent TRPM8 antagonists within their respective series. For our docking studies, we employed software tools in the Yasara suite of programs, and the interactions were analyzed with Yasara and Protein–Ligand Interaction Profiler (PLIP). Docking simulations were performed on a model of the rat TRPM8 channel generated from the cryoelectron microscopy (cryo-EM) structure of *Ficedula albicollis* TRPM8 (PDB code 6BPQ) [54]. 

Initially, we used the TRPM8 structure encompassing transmembrane helices as well as the internal and extracellular loops of the indicated tetrameric channel. As outlined in Table 6, the most prominent binding solutions for the three compounds consistently aligned them within or close to the pore channel. Curiously, these calculations did not provided any solution in which compounds **34**, **35**, or **37** bind to the menthol-binding pocket (subsite 5) [42]. A few solutions, generally with decreased binding energy, positioned these compounds lower in the menthol binding site, in between the low part of S1–S4 and the TRP domain (subsite 6).

Upon analyzing the most populated solutions for the new described compounds, we identified four primary binding subsites within the TRPM8 tetrameric protein (see Appendix A for the location of the different subsites). Progressing from the top (extracellular) to the bottom (intracellular) regions, we observed that 14 to 28% of the solutions were clustered around the tower formed by the external loops (Subsite 1), being the second subsite most populated and that of the better biding energy in the case of monobenzyl derivative **37**. A second subsite (subsite 2), involving transmembrane helices S3 and S4 of a protein subunit and S6 of an adjacent monomer, was found to be more populated for the dibenzyl amide derivatives **34** and **35** in comparison to the monobenzyl analogue **37**. Subsite 3, bounded by interactions with transmembrane helices S5 and S6 from a single monomer, along with certain loops near the pore helix on the same monomer and a neighboring protein subunit, emerged as the most populated binding site for all three compounds. Subsite 4, located within the cytoplasmic pore loops, accumulated a greater number of solutions for the smaller monobenzyl derivative **37** when compared to the dibenzyl analogues **34** and **35**. However, for the latter compounds, subsite 4 corresponded to the binding pocket with the better binding energy. A comparative simulation using menthol as a ligand recognized subsites 1 and 2, but not 3 and 4, with an important population within the menthol-binding site (subsite 5) and the surrounding subsite 6. However, simulation studies with AMTB showed a similar behavior to those of **34**, **35** and **37** with only minor location of the antagonist at the menthol cavity and approximately 11% location at subsite 6. The recently resolved cryo-EM structures of TRPM8 complexed with known antagonists, AMTB and TC-I2014, showed that these compounds were situated within a well-defined pocket related to subsite 6 that was formed by the lower part of the TM S1–S4 helices and the TRP domain [42,55]. As TRPM8 mediated the permeation of large cations across cell membranes [56], and both AMTB and our β-lactamas are cationic molecules, the preferential situation of these molecules by the pore hole could be plausible.

To investigate whether the extracellular loops in the previous model play a role in guiding the distribution of β-lactam derivatives into cavities along the pore zone, as previously hypothesized for related analogs [35], an additional structure of the channel was incorporated into the modeling studies (compounds **34** and **37)**. This alternate structure retained the transmembrane helices and inner pore loops but excluded the extracellular loops. Within this framework, many different solutions emerged, but subsites 2, 3 and 4 remained the most frequently populated solutions with an increased preference for subsite 4 in the case of the smaller compound **37** (see Appendix A for details). Once more, no interactions with the menthol binding site were observed.

The stabilizing interactions among the external loops of the channel (subsite 1) and amide derivatives **34** and **35** are characterized by hydrophobic contacts with L889, V903, V919, T922, T923, F926, K937, and L939, among others, from a single protein monomer (Figure 5A). Using the PLIP program, both compounds establish a π–cation interconnection, between R890 and the Ph ring of the 4-Bn group in **34**, and between K937 and one of the aromatic rings of the N-Bn_2_ moiety in **35**. Calculations conducted with Yasara identified some π-π interactions involving F900 and/or F926. Furthermore, both software programs indicate that the CO group of the β-lactam ring in **35** engages in a hydrogen bond with NH/NH_2_ protons of the R890 guanidino group.

Within subsite 2 (Figure 4B), amides **34** and **35** exhibit two π-π interactions involving F815 and W954 of the protein and different phenyl groups of the small-molecule. Additionally, different hydrophobic contacts are observed, which are mainly delimited by Y808 (TM S3) and F815, I832 and I831 (TM S4) of a monomer as well as by P949 and F951 of TM S6 of an adjacent protein unit (Figure 5B).

Deeper within the pore channel, subsite 3 corresponds to the most populated docking solutions for both compounds **34** and **35**. This subsite entails two transmembrane helices from a protein subunit, namely S5 and S6, and two loops positioned at the exit of the pore helix: one from the same monomer and the other from a contiguous protein in the tetramer and its S5. At this subsite, both software programs identify an H-bond involving the α-CO of G913 residue, acting as the acceptor, while the 4-CONH amide donates its proton to the interaction (identified in both compounds, **34** and **35**). In addition, several hydrophobic contacts occur among different parts of the indicated small-molecules (NBn_2_, 2′-Bn, 3-Me, and *^i^*Bu or Bn amide group) and different protein residues. These include interactions with L909 and L915 of the S5-S6 connector, at the end of pore helix (two subunits), and L959, V960, I962 and Y963 of transmembrane S6 (Figure 5C). Residues W877 or F874 from S5 are also implicated in the complex stabilizing interactions.

At the pore internal mouth (Subsite 4), amide compounds **34** and **35** interconnect with residues spanning all four protein units. The hydrophobic interactions are primarily enabled by Y981 (4 subunits), I985 (3), T982 (1) and M978 (1) residues. However, both compounds adopt different poses within the channel and establish a singular H-bond between the α-CO of V983 and the 4-CONH proton in **34** (Figure 5D), while the 4-CO of the β-lactam connects with the α-NH proton of residue I985 in **35**. Additionally, the aromatic side chains of Tyr981 residues are able to engage in π-π stacking and T-edge interactions with the aromatic rings of either 2′-Bn and N-Bn_2_ in **34** or with that of 4-Bn in **35**.

The monobenzyl derivative **37** has binding interactions with the same four subsites as its dibenzyl analogue **34**. However, the population of solutions is increased at subsites 1, 3 and 4. Residues of the protein implicated in the hydrophobic interconnections with **37** are similar to those above described for the dibenzyl compound at pockets 2–4. However, a different pose was identified at subsite 1, where Y924 residues from all four protein subunits enclose the molecules with the assistance of D920 from a single monomer. Moreover, a π-π stacking interaction is formed between one of these Y924 residues and the phenyl ring at 2′-position of **37**. This compound also engages in two H-bonds: one involving the 2′-NH and the CO of T982 and the other involving the CO of the β-lactam ring and the α-NH of one of the Y981 residues.

Regarding subsite 6, located between the TRP domain and the lower part of the S1 and S4 TM domains, main residues involved in AMTB interactions were P734, F735, F738, S739, L853, L1001, V1002, E1004, Y1005 and L1009. The mono- and dibenzyl derivatives established additional contacts in this subsite, thus strongly embracing the S4 TM and TRP domains. These new interactions observed were hydrophobic (F847) or polar (S850), including a hydrogen bond with derivative **37** or even additional interactions in the TRP domain (Y999, R998) for derivative **35**.

In general, the interconnexion of these β-lactam derivatives at the main four TRPM8 subsites are mostly hydrophobic or involve aromatic π interconnections, which are in agreement with the lower experimental activity obtained for compounds including extra polar nitrogens on the amide moiety. A similar explanation could be given for the reduced activity of monobenzyl derivatives bearing CN and OR substituents, while hydrophobic moieties (Me, Ph, Br, Nf) maintained or increased the antagonist activity. It is interesting to note that the CO molecules of the β-lactam ring of **35** and **37** are involved in a H-bond with protein NHs at some subsites. The formation of H-bonds between the NH of the 2-amide moiety, as for compound **35** in subsite 1 and **34** in subsite 3, and the channel could be behind the improved activity of amides compared to the ester model compound.

Although the main docking solutions for β-lactam derivatives indicated a high probability for these molecules to align by the exe of the pore channel, a similar location to that described for AMTB using Cryo-EM techniques cannot be discarded, since this cavity has been revealed to be a big and adaptable pocket [55].

## 3. Materials and Methods

### 3.1. Synthesis

General procedures, synthesis of key intermediates and specific details on the characterization of diastereomeric β-lactams are gathered in the Appendix A.

General method for the synthesis of substituted 4-carboxamides. To a solution of the β-lactam 4-carboxylate (0.188 mmol) in dry DCM (5 mL) was added PyBrOP (0.225 mmol, 0.105 g), TEA (0.225 mmol, 0.031 mL) and the corresponding amine (0.225 mmol). The reaction was stirred at rt, and after the starting product disappeared, the solvent was evaporated to dryness. The reaction mixture was dissolved in EtOAc, washed with 0.1 M HCl, NaHCO_3_ (10%) and saturated NaCl solution. The organic phase was dried over anhydrous Na_2_SO_4_, filtered and evaporated to dryness. The resulting residue was purified on a silica gel column, using the eluent system indicated in each case.

General procedure for the synthesis of 2′-N-monobenzyl derivatives. To a solution of the corresponding 2′-NH_2_ β-lactam derivative (0.225 mmol) in MeOH (4 mL) was added TEA (0.225 mmol, 0.031 mL) and the corresponding aldehyde (0.337 mmol). The reaction mixture was stirred for 1.5 h at rt. Then, NaBH_4_ (0.450 mmol, 0.017 g) was added at 0 °C and stirred at rt for additional 24 h, and finally, the solvent was evaporated to dryness. The organic residue was dissolved in EtOAc and washed with H_2_O and saturated NaCl solution successively. The organic phase was dried over anhydrous Na_2_SO_4_, filtered and evaporated to dryness. The resulting residue was purified on a silica gel column, using the eluent system indicated in each case.

4*S*-Benzyl-4-[*N-(iso-*butyl)-carbamoyl]-3*S*-methyl-1-[(2′*S*-dibenzylamino-3′-phenyl)prop-1′-yl]-2-oxoazetidine (**34**). Syrup. Yield: 86% (from **33** and *iso*-butylamine). Eluent: 20 to 30% of EtOAc in hexane. HPLC: t_R_ = 7.54 min (gradient from 15% to 95% of A in 10 min). [α]_D_ = −25.50 (c 1, CH_3_Cl). ^1^H-NMR (400 MHz, CDCl_3_): *δ* 7.75–6.89 (m, 20H, Ar), 6.08 (t, *J =* 6.5 Hz, 1H, NH), 3.72 (s, 4H, NCH_2_), 3.72 (dd, *J =* 14.3, 6.7 Hz, 1H, H_1′_), 3.25 (p, *J =* 6.9 Hz, 1H, H_2′_), 3.18 (dd, *J =* 14.3, 3.8 Hz, 1H, H_1′_), 3.16 (d, *J =* 14.5 Hz, 1H, 4-CH_2_), 3.16 (m, 1H, H_3_), 3.02 (d, *J =* 14.6 Hz, 1H, 4-CH_2_), 2.95 (dd, *J =* 13.3, 6.7 Hz, 1H, CH_2_, *^i^*Bu), 2.84 (dd, *J =* 13.8, 6.6 Hz, 1H, H_3′_), 2.82 (dd, *J =* 13.4, 6.6 Hz, 1H, CH_2_, *^i^*Bu), 2.61 (dd, *J =* 13.8, 7.1 Hz, 1H, H_3′_), 1.46 (m, 1H, CH, *^i^*Bu), 1.18 (d, *J =* 7.5 Hz, 3H, 3-CH_3_), 0.75 (d, *J =* 6.7 Hz, 3H, CH_3_, *^i^*Bu), 0.71 (d, *J =* 6.7 Hz, 3H, CH_3_, *^i^*Bu). ^13^C-NMR (75 MHz, CDCl_3_): *δ* 171.7 (C_2_), 170.3 (4-CONH), 139.5, 139.2, 135.9, 130.0, 129.3, 129.1, 129.0, 128.5, 128.5, 127.4, 127.3, 126.3 (Ar), 69.4 (C_4_), 60.0 (C_2′_), 54.0 (C_3_), 53.2 (NCH_2_), 47.4 (CH_2_, *^i^*Bu), 42.8 (C_1′_), 40.2 (4-CH_2_), 36.0 (C_3′_), 28.4 (CH, *^i^*Bu), 20.4 (CH_3_, *^i^*Bu), 20.3 (CH_3_, *^i^*Bu), 10.4 (3-CH_3_). MS(ES)^+^: 588.51 [M + H]^+^. Exact mass calculated for C_39_H_45_N_3_O_2_: 587.35118, found 587.35311.

4*S*-Benzyl-4-[*N*-(benzyl)-carbamoyl]-3*S*-methyl-1-[(2′*S*-dibenzylamino-3′-phenyl)prop-1′-yl]-2-oxoazetidine (**35**). Syrup. Yield: 74% (from **33** and benzylamine). Eluent: 20% to 30% of EtOAc in hexane. HPLC: t_R_ = 7.60 min (gradient from 15% to 95% of A in 10 min). [α]_D_ = −41.10 (c 1, CH_3_Cl). ^1^H-NMR (400 MHz, CDCl_3_): *δ* 7.45 (m, 2H, Ar, NH), 7.31–7.13 (m, 18H, Ar), 7.09 (m, 2H, Ar), 7.01 (m, 2H, Ar), 6.89 (m, 2H, Ar), 4.53 (dd, *J =* 14.6, 6.4 Hz, 1H, NHC*H*_2_, Bn), 4.15 (dd, *J =* 14.6, 6.3 Hz, 1H, NHC*H*_2_, Bn), 3.77 (m, 1H, H_1′_), 3.64 (d, *J =* 14.4 Hz, 2H, NCH_2_), 3.58 (d, *J =* 14.4 Hz, 2H, NCH_2_), 3.37 (d, *J =* 14.6 Hz, 1H, 4-CH_2_), 3.23 (q, *J =* 7.3 Hz, 1H, H_3_), 3.17 (m, 1H, H_1′_), 3.14 (m, 1H, H_2′_), 3.11 (d, *J =* 14.7 Hz, 1H, 4-CH_2_), 2.66 (dd, *J =* 13.7, 5.6 Hz, 1H, H_3′_), 2.44 (dd, *J =* 13.6, 8.2 Hz, 1H, H_3′_), 1.23 (d, *J =* 7.5 Hz, 3H, 3-CH_3_). ^13^C-NMR (75 MHz, CDCl_3_): *δ* 172.2 (C_2_), 170.7 (4-CONH), 139.2, 138.6, 138.1, 135.6, 130.0, 129.2, 129.1, 128.8, 128.7, 128.6, 128.5, 128.1, 127.6, 127.4, 127.3, 126.3 (Ar), 69.7 (C_4_), 60.3 (C_2′_), 54.5 (C_3_), 52.8 (NCH_2_), 43.9 (NHCH_2_, Bn), 42.6 (C_1′_), 39.8 (4-CH_2_), 35.9 (C_3′_), 10.2 (3-CH_3_). MS(ES)^+^: 622.50 [M + H]^+^. Exact mass calculated for C_42_H_43_N_3_O_2_: 621.33553, found 621.33767.

4*S*-Benzyl-4-[*N*-(3′′-pyridyl)methylcarbamoyl]-3*S*-methyl-1-[(2′*S*-dibenzylamino-3′-phenyl)prop-1′-yl]-2-oxoazetidine (**36**). Syrup. Yield: 73% (from **33** and 3-picoline). Eluent: 10% of EtOAc in DCM. HPLC: t_R_ = 5.93 min (gradient from 15% to 95% of A in 10 min). [α]_D_ = −39.71 (c 1, CH_3_Cl). ^1^H-NMR (400 MHz, CDCl_3_): *δ* 8.44 (dd, *J =* 4.9, 1.7 Hz, 1H, H_4′′_), 8.22 (d, *J =* 2.3 Hz, 1H, H_2′′_), 7.64 (t, *J =* 5.9 Hz, 1H, NH), 7.33 (dt, *J =* 7.8, 2.0 Hz, 1H, H_6′′_), 7.23–7.00 (m, 16H, H_5′′_, Ar), 6.86 (m, 2H, Ar), 6.75 (m, 2H, Ar), 6.60 (m, 1H, Ar), 4.32 (dd, *J =* 14.7, 6.1 Hz, 1H, NHC*H*_2_), 4.08 (dd, *J =* 14.8, 5.8 Hz, 1H, NHC*H*_2_), 3.88 (d, *J =* 13.8 Hz, 2H, NCH_2_), 3.88 (m, 1H, H_1′_), 3.75 (d, *J =* 13.9 Hz, 2H, NCH_2_), 3.37 (d, *J =* 14.7 Hz, 1H, 4-CH_2_), 3.23 (q, *J =* 7.4 Hz, 1H, H_3_), 3.13 (d, *J =* 14.6 Hz, 1H, 4-CH_2_), 3.12 (m, 2H, H_1′_, H_2′_), 2.75 (dd, *J =* 13.6, 4.3 Hz, 1H, H_3′_), 2.40 (dd, *J =* 13.5, 8.6 Hz, 1H, H_3′_), 1.20 (d, *J =* 7.6 Hz, 3H, 3-CH_3_). ^13^C-NMR (75 MHz, CDCl_3_): *δ* 172.2 (C_2_), 171.3 (4-CONH), 149.6, 149.0, 139.0, 138.5, 136.1, 135.5, 133.8, 130.0, 129.4, 129.0, 128.9, 128.7, 128.7, 127.6, 127.3, 126.5, 123.5 (Ar), 69.7 (C_4_), 60.7(C_2′_), 54.7 (C_3_), 53.0 (NCH_2_), 42.4 (C_1′_), 41.4 (NHCH_2_), 39.4 (4-CH_2_), 35.5 (C_3′_), 10.2 (3-CH_3_). MS(ES)^+^: 623.44 [M + H]^+^. Exact mass calculated for C_41_H_42_N_4_O_2_: 622.33078, found 622.33184.

4*S*-Benzyl-4-(*terc*-butoxy)carbonyl-3*S*-methyl-1-[(2’*S-N-*benzylamino-3’-phenyl) prop-1′-yl]-2-oxoazetidine (**37**). Syrup. Yield: 10% (from **1**). Eluent: 3% to 9% of MeOH in DCM. HPLC: t_R_ = 4.74 min (gradient from 15% to 95% of A in 5 min). [α]_D_ = +25.50 (c 1, CH_3_Cl). ^1^H-NMR (400 MHz, CDCl_3_): *δ* 7.34–7.20 (m, 11H, Ar), 7.14 (m, 4H, Ar), 3.83 (d, *J =* 13.2 Hz, 1H, NHCH_2_), 3.76 (d, *J =* 13.2 Hz, 1H, NHCH_2_), 3.44 (d, *J =* 14.5 Hz, 1H, 4-CH_2_), 3.30 (m, 1H, H_2′_), 3.20 (dd, *J =* 14.1, 7.8 Hz, 1H, H_1′_), 3.09 (q, *J =* 7.7 Hz, 1H, H_3_), 3.07 (d, *J =* 14.5 Hz, 1H, 4-CH_2_), 3.02 (dd, *J =* 14.1, 4.6 Hz, 1H, H_1′_), 2.84 (dd, *J =* 13.9, 6.1 Hz, 1H, H_3′_), 2.75 (dd, *J =* 13.9, 6.9 Hz, 1H, H_3′_), 2.33 (s ancho, 1H, NH), 1.39 (s, 9H, CH_3_, *^t^*Bu), 1.21 (d, *J =* 7.5 Hz, 3H, 3-CH_3_). ^13^C-NMR (75 MHz, CDCl_3_): *δ* 170.6 (COO), 170.1 (C_2_), 138.7, 135.5, 130.1, 129.5, 128.7, 128.5, 128.5, 128.5, 127.4, 127.0, 126.4 (Ar), 83.2 (C, *^t^*Bu), 68.5 (C_4_), 57.7 (C_2′_), 53.4 (C_3_), 51.3 (NCH_2_), 47.4 (C_1’_), 41.0 (4-CH_2_), 39.1 (C_3’_), 28.1 (CH_3_, *^t^*Bu), 10.6 (3-CH_3_). MS(ES)^+^: 499.42 [M + H]^+^. Exact mass calculated for C_38_H_42_N_2_O_3_: 498.2882, found 498.2881.

4*S*-Benzyl-4-(*terc*-butoxy)carbonyl-3*S*-methyl-1-[(2’*S-N-*(3′′-biphenyl)methyl amino-3’-phenyl) prop-1′-yl]-2-oxoazetidine (**40**). Syrup. Yield: 77% (from 38 and biphenyl-3-carboxaldehyde). Eluent: 9% to 100% of EtOAc in hexane. HPLC: t_R_ = 7.43 min (gradient from 15% to 95% of A in 10 min). [α]_D_ = +25.50 (c 1, CH_3_Cl). ^1^H-NMR (400 MHz, CDCl_3_): *δ* 7.56 (m, 2H, Ar), 7.43 (m, 4H, Ar), 7.33 (m, 2H, Ar), 7.21 (m, 7H, Ar), 7.11 (m, 4H, Ar), 3.82 (d, *J =* 13.2 Hz, 1H, NHCH_2_), 3.77 (d, *J =* 13.2 Hz, 1H, NHCH_2_), 3.40 (d, *J =* 14.5 Hz, 1H, 4-CH_2_), 3.28 (m, 1H, H_2′_), 3.15 (q, *J =* 7.6 Hz, 1H, H_3_), 3.05 (d, *J =* 14.5 Hz, 1H, 4-CH_2_), 3.04 (m, 2H, H_1′_), 2.78 (dd, *J =* 13.9, 6.5 Hz, 1H, H_3′_), 2.71 (dd, *J =* 13.9, 6.7 Hz, 1H, H_3′_), 1.66 (s ancho, 1H, NH), 1.34 (s, 9H, CH_3_, *^t^*Bu), 1.14 (d, *J =* 7.5 Hz, 3H, 3-CH_3_). ^13^C-NMR (75 MHz, CDCl_3_): *δ* 170.5 (COO), 170.2 (C_2_), 141.3, 139.0, 135.6, 130.1, 129.5, 128.9, 128.8, 128.7, 128.5, 127.4, 127.3, 127.3, 127.3, 127.2, 126.3, 125.7 (Ar), 83.2 (C, *^t^*Bu), 68.5 (C_4_), 57.6 (C_2′_), 53.5 (C_3_), 51.5 (NCH_2_), 47.6 (C_1’_), 41.0 (4-CH_2_), 39.5 (C_3’_), 28.1 (CH_3_, *^t^*Bu), 10.6 (3-CH_3_). MS(ES)^+^: 575.36 [M + H]^+^. Exact mass calculated for C_38_H_42_N_2_O_3_: 574.31954, found 574.31961.

4*S*-Benzyl-4-*terc*-butoxycarbonyl-3*S*-methyl-1-[(2’*S-N-*(4′′-biphenyl)methylamino-3’-phenyl)prop-1′-yl]-2-oxoazetidine (**41**). Syrup. Yield: 68% (from **38** and 4-biphenyl-carboxaldehyde). Eluent: 9% to 100% of EtOAc in hexane. HPLC: t_R_ = 7.44 min (gradient from 15% to 95% of A in 10 min). [α]_D_ = −17.99 (c 1, CH_3_Cl). ^1^H-NMR (400 MHz, CDCl_3_): *δ* 7.56 (m, 2H, Ar), 7.50 (m, 2H, Ar), 7.42 (m, (2H, Ar), 7.35–7.18 (m, 9H, Ar), 7.12 (m, 4H, Ar), 3.80 (d, *J =* 13.2 Hz, 1H, NHCH_2_), 3.74 (d, *J =* 13.1 Hz, 1H, NHCH_2_), 3.41 (d, *J =* 14.5 Hz, 1H, 4-CH_2_), 3.28 (m, 1H, H_2′_), 3.15 (q, *J =* 7.6 Hz, 1H, H_3_), 3.06 (d, *J =* 14.6 Hz, 1H, 4-CH_2_), 3.05 (m, 2H, H_1′_), 2.79 (dd, *J =* 13.9, 6.5 Hz, 1H, H_3′_), 2.71 (dd, *J =* 13.9, 6.7 Hz, 1H, H_3′_), 1.64 (s ancho, 1H, NH), 1.36 (s, 9H, CH_3_, *^t^*Bu), 1.19 (d, *J =* 7.5 Hz, 3H, 3-CH_3_). ^13^C-NMR (75 MHz, CDCl_3_): *δ* 170.6 (COO), 170.2 (C_2_), 141.3, 134.0, 139.8, 139.0, 135.6, 130.1, 129.6, 129.5, 128.8, 128.8, 128.7, 128.5, 127.4, 127.2, 127.2, 126.3 (Ar), 83.2 (C, *^t^*Bu), 68.5 (C_4_), 57.7 (C_2′_), 53.5 (C_3_), 51.1 (NCH_2_), 47.7 (C_1′_), 41.02 (4-CH_2_), 39.6 (C_3’_), 28.2 (CH_3_, *^t^*Bu), 10.7 (3-CH_3_). MS(ES)^+^: 575.28 [M + H]^+^. Exact mass calculated for C_38_H_42_N_2_O_3_: 574.31954, found 574.32059.

4*S*-Benzyl-4-[*N-(iso-*butyl)-carbamoyl]-3*S*-methyl-1-[(2′*S-*(3′′-phenylbencil)amino-3′-phenyl)prop-1′-yl]-2-oxoazetidine (**42**). Syrup. Yield: 40% (from 39 and 3-phenylbenzaldehyde). Eluent: 25% of EtOAc in hexane. HPLC: t_R_ = 7.35 min (gradient from 15% to 95% of A in 10 min). [α]_D_ = −82.17 (c 1, CH_3_Cl). ^1^H-NMR (300 MHz, CDCl_3_): *δ* 10.22 (t, *J =* 5.5 Hz, 1H, 4-CONH), 7.47 (m, 5H, Ar), 7.42–7.21 (m, 10H, Ar), 7.02 (m, 3H, Ar), 6.78 (dt, *J =* 7.6, 1.4 Hz, 1H, Ar), 4.07 (d, *J =* 14.2 Hz, 1H, 4-CH_2_), 3.64 (dd, *J =* 14.9, 6.3 Hz, 1H, H_1′_), 3.40 (d, *J =* 12.8 Hz, 1H, NHCH_2_), 3.32 (d, *J =* 12.8 Hz, 1H, NHCH_2_), 3.14 (q, *J =* 7.6 Hz, 1H, H_3_), 3.08 (m, 2H, CH_2_, *^i^*Bu, H_1′_), 2.98 (d, *J =* 14.3 Hz, 1H, 4-CH_2_), 2.73 (dd, *J =* 12.6, 2.6 Hz, 1H, H_3′_), 2.67–2.50 (m, 3H, H_2′_, H_3′_, NHC*H*_2_), 1.62 (s ancho, 1H, NH), 1.58 (m, 2H, NH, CH, *^i^*Bu), 1.26 (d, *J =* 7.5 Hz, 3H, 3-CH_3_), 0.70 (d, *J =* 6.6 Hz, 3H, CH_3_, *^i^*Bu), 0.69 (d, *J =* 6.6 Hz, 3H, CH_3_, *^i^*Bu_′_). ^13^C-NMR (75 MHz, CDCl_3_): *δ* 173.5 (4-CONH), 170.8 (C_2_), 141.7, 140.8, 139.0, 137.9, 137.0, 130.3, 129.1, 129.0, 129.0, 128.9, 128.8, 127.6, 127.4, 127.2, 126.9, 126.9, 126.9, 126.2 (Ar), 70.6 (C_4_), 56.3 (C_2′_), 55.8 (C_3_), 47.8 (NCH_2_), 47.1 (CH_2_, *^i^*Bu), 42.7 (C_1′_), 41.7 (4-CH_2_), 38.3 (C_3′_), 28.1 (CH, *^i^*Bu), 20.4 (CH_3_, *^i^*Bu), 10.9 (3-CH_3_). MS(ES)^+^: 574.58 [M + H]^+^. Exact mass calculated for C_38_H_43_N_3_O_2_: 573.33553, found 573.33686.

4*S*-Benzyl-4-[*N-(iso-*butyl)carbamoyl]-3*S*-methyl-1-[(2′*S-*(3′′-fluorobenzyl)amino-3′-phenyl)prop-1′-yl]-2-oxoazetidine (**43**). Syrup. Yield: 53% (from **39** and 3-fluorobenzaldehyde). Eluent: 25% of EtOAc in hexane. HPLC: t_R_ = 6.82 min (gradient from 15% to 95% of A in 10 min). [α]_D_ = −42.44 (c 1, CH_3_Cl). ^1^H-NMR (300 MHz, CDCl_3_): *δ* 9.99 (s ancho, 1H, 4-CONH), 7.30 (m, 8H, Ar), 7.16 (m, 1H, Ar), 6.96 (m, 2H, Ar), 6.88 (m, 1H, Ar), 6.58 (d, *J =* 7.6 Hz, 1H, Ar), 6.48 (d, *J =* 9.7 Hz, 1H, Ar), 4.06 (d, *J =* 14.2 Hz, 1H, 4-CH_2_), 3.61 (dd, *J =* 14.9, 6.5 Hz, 1H, H_1′_), 3.36 (d, *J =* 13.0 Hz, 1H, NHCH_2_), 3.29 (d, *J =* 13.1 Hz, 1H, NHCH_2_), 3.14 (q, *J =* 7.2 Hz, 1H, H_3_), 3.12 (m, 2H, H_1′_, CH_2_, *^i^*Bu), 2.97 (d, *J =* 14.3 Hz, 1H, 4-CH_2_), 2.72–2.49 (m, 4H, H_2′_, CH_2_, *^i^*Bu,H_3′_), 1.68 (s ancho, 1H, NH), 1.60 (m, 2H, NH, CH, *^i^*Bu), 1.26 (d, *J =* 7.4 Hz, 3H, 3-CH_3_), 0.74 (d, *J =* 6.7 Hz, 3H, CH_3_, *^i^*Bu), 0.73 (d, *J =* 6.7 Hz, 3H, CH_3_, *^i^*Bu). ^13^C-NMR (75 MHz, CDCl_3_): *δ* 173.4 (4-CONH), 170.7 (C_2_), 162.9 (d, *J =* 24.8 Hz, C_3′′_), 141.0 (d, *J =* 7.3 Hz, C_1′′_), 137.7, 136.9, 130.3, 130.2 (d, *J =* 8.7 Hz, C_5′′_), 129.0, 128.9, 128.8, 127.3, 127.0, 123.6 (d, *J =* 3.0 Hz, C_6′′_), 114.9 (d, *J =* 21.4 Hz, C_2′′_), 114.4 (d, *J =* 21.0 Hz, C_4′′_) (Ar), 70.5 (C_4_), 56.1 (C_2′_), 55.8 (C_3_), 47.2 (CH_2_, *^i^*Bu), 47.0 (NCH_2_), 42.6 (C_1′_), 41.6 (4-CH_2_), 38.1 (C_3′_), 28.2 (CH, *^i^*Bu), 20.4 (CH_3_, *^i^*Bu), 10.7 (3-CH_3_). MS(ES)^+^: 516.55 [M + H]^+^, 518.54 [M + 2]^+^. Exact mass calculated for C_32_H_38_FN_3_O_2_: 515.29481, found 515.2949.

4*S*-Benzyl-4-[*N-(iso-*butyl)carbamoyl]-3*S*-methyl-1-[(2′*S-*(3′′-bromobenzyl)amino-3′-phenyl)prop-1′-yl]-2-oxoazetidine (**44**). Syrup. Yield: 43% (from **39** and 3-bromobenzaldehyde). Eluent: 9% to 14% of EtOAc in DCM. HPLC: t_R_ = 7.04 min (gradient from 15% to 95% of A in 10 min). [α]_D_ = −88.58 (c 1, CH_3_Cl). ^1^H-NMR (300 MHz, CDCl_3_): *δ* 9.98 (t, *J =* 5.4 Hz, 1H, 4-CONH), 7.37–7.19 (m, 8H, Ar), 7.06 (t, *J =* 7.8 Hz, 1H, Ar), 7.00–6.91 (m, 4H, Ar), 6.70 (dt, *J =* 7.6, 1.3 Hz, 1H, Ar), 4.08 (d, *J =* 14.3 Hz, 1H, 4-CH_2_), 3.61 (dd, *J =* 14.9, 6.6 Hz, 1H, H_1′_), 3.32 (d, *J =* 13.1 Hz, 1H, NHCH_2_), 3.24 (d, *J =* 13.0 Hz, 1H, NHCH_2_), 3.14 (q, *J =* 7.3 Hz, 1H, H_3_), 3.13 (m, 1H, CH_2_, *^i^*Bu), 3.03 (dd, *J =* 15.1, 2.8 Hz, 1H, H_1′_), 2.96 (d, *J =* 14.3 Hz, 1H, 4-CH_2_), 2.74–2.46 (m, 4H, H_2′_, H_3′_, CH_2_, *^i^*Bu), 1.58 (m, 2H, NH, CH, *^i^*Bu), 1.55(s ancho, 1H, NH), 1.27 (d, *J =* 7.5 Hz, 3H, 3-CH_3_), 0.75 (d, *J =* 6.6 Hz, 3H, CH_3_, *^i^*Bu), 0.75 (d, *J =* 6.7 Hz, 3H, CH_3_, *^i^*Bu). ^13^C-NMR (75 MHz, CDCl_3_): *δ* 173.4 (4-CONH), 170.7 (C_2_), 140.9, 137.7, 137.0, 131.2, 130.6, 130.3, 130.2, 129.1, 128.9, 128.9, 127.4, 127.1, 126.6, 122.8 (Ar), 70.6 (C_4_), 56.0 (C_2′_), 56.0 (C_3_), 47.2 (CH_2_, *^i^*Bu), 46.9 (NCH_2_), 42.5 (C_1′_), 41.7 (4-CH_2_), 38.2 (C_3′_), 28.2 (CH, *^i^*Bu), 20.5 (CH_3_, *^i^*Bu), 10.8 (3-CH_3_). MS(ES)^+^: 576.49 [M + H]^+^, 578.56 [M + 2]^+^. Exact mass calculated for C_32_H_38_BrN_3_O_2_: 575.21474, found 575.21447.

4*S*-Benzyl-4-[*N-(iso-*butyl)-carbamoyl]-3*S*-methyl-1-[(2′*S-*(2′′*-*naphtylmethyl) amino-3′-phenyl)prop-1′-yl]-2-oxoazetidine (**45**). Syrup. Yield: 41% (From **39** and 2-naphtaldehyde). Eluent: 33% of EtOAc in DCM. HPLC: t_R_ = 7.35 min (gradient from 15% to 95% of A in 10 min). [α]_D_ = −62.25 (c 1, CH_3_Cl). ^1^H-NMR (300 MHz, CDCl_3_): *δ* 10.10 (s ancho, 1H, 4-CONH), 7.78 (d, *J =* 8.1 Hz, 1H, Ar), 7.71 (d, *J =* 8.3 Hz, 1H, Ar), 7.43–7.19 (m, 12H, Ar), 6.97 (m, 3H, Ar), 4.11 (d, *J =* 14.2 Hz, 1H, 4-CH_2_), 3.85 (d, *J =* 11.8 Hz, 1H, NHC*H*_2_), 3.79 (dd, *J =* 14.9, 6.9 Hz, 1H, H_1′_), 3.64 (d, *J =* 12.0 Hz, 1H, NHC*H*_2_), 3.17 (q, *J =* 7.2 Hz, 1H, H_3_) 3.15 (m, 1H, H_1′_), 3.00 (d, *J =* 14.2 Hz, 1H, 4-CH_2_), 2.79–2.52 (m, 4H, H_3′_, CH_2_, *^i^*Bu, H_2′_), 2.31 (m, 1H, CH_2_, *^i^*Bu), 1.50 (s ancho, 1H, NH), 1.28 (d, *J =* 7.5 Hz, 3H, 3-CH_3_), 1.13 (m, 1H, CH, *^i^*Bu), 0.45 (d, *J =* 6.6 Hz, 3H, CH_3_, *^i^*Bu), 0.40 (d, *J =* 6.6 Hz, 3H, CH_3_, *^i^*Bu). ^13^C-NMR (75 MHz, CDCl_3_): *δ* 173.6 (4-CONH), 170.6 (C_2_), 137.8, 137.1, 134.3, 133.8, 131.5, 130.4, 129.0, 128.8, 128.3, 127.4, 127.0, 126.8, 126.5, 125.8, 125.4, 122.8 (Ar), 70.8 (C_4_), 57.0 (C_2′_), 56.0 (C_3_), 46.9 (CH_2_, *^i^*Bu), 45.5 (NHCH_2_), 42.9 (C_1′_), 41.8 (4-CH_2_), 38.0 (C_3′_), 27.7 (CH, *^i^*Bu), 20.2 (CH_3_, *^i^*Bu), 10.8 (3-CH_3_). MS(ES)^+^: 548.63 [M + H]^+^. Exact mass calculated for C_36_H_41_N_3_O_2_: 547.31988, found 547.32163.

### 3.2. Molecular Modeling

The docking protocol used the algorithm AutoDockLGA [57] and the force field AMBER03 [58] implemented in Yasara software version 22.5.22 [59]. Interaction analysis was completed with PLIP, which is a fully automated protein–ligand interaction server [60]. Figures were drawn with PyMol (The PyMOL Molecular Graphics System, Version 2.6 Schrödinger, LLC., New York, NY, USA). Retrieved from http://www.pymol.org/pymol (accessed 25 August 2023). Detailed interactions can be seen in the Appendix A.

### 3.3. Biological Assays

Calcium fluorometry, Patch-Clamp assays and in vivo experiments were performed using the methodology previously described for TRPM8 [35] or TRPM3 [61]. In the Patch -Clamp experiments, the intracellular pipette solution contained (in mM) 140 KCl, 5 EGTA, and 10 mM HEPES, adjusted to pH 7.2 with KOH, and the extracellular solution contained (in mM) 140 NaCl, 2 CaCl_2_, 1 MgCl_2_, 10 D-glucose and 10 HEPES, adjusted to pH 7.4 with NaOH. Selectivity against TRPV1, TRPV3, TRPA1 and ASIC3 was subcontracted to Eurofins-CEREP or Eurofins-PANLABS (see ref. [35] for more details).

## 4. Conclusions

Two novel series of β-lactam derivatives, featuring *N*-substituted 4-carboxamides and/or 2′-*N*′-monobenzyl groups, have been prepared and analyzed regarding their capacity to inhibit the Ca^2+^ entry triggered by menthol in TRPM8 channels. Structure–activity relationships unveiled that *N*′-monobenzyl derivatives, regardless of the size and the electronic character of the substituents on the phenyl ring, failed to compete with the antagonist activity demonstrated by the corresponding *N*′,*N*′-dibenzyl analogues. However, the 4-isobutyl- and 4-benzylcarboxamides, having a minute increase of 3 units in the TPSA, exhibited enhanced inhibitory potency against the agonist-induced TRPM8 activation when contrasted with the O*^t^*Bu analogue. Unfortunately, the introduction of an extra nitrogen atom within the *N*-benzyl moiety, leading to increased TPSA values, visibly compromised the activity. The most potent compounds within the 4-carboxamide series, **34** and **35**, confirmed their activity in electrophysiology assays, showed selectivity for the TRPM8 channel, and displayed antiallodynic activity in a model of oxaliplatin-induced peripheral neuropathy. Ongoing research within our group to further explore this family of TRPM8 antagonists is directed to enhance TPSA values while maintaining the antagonist activity.

## Data Availability

Most data from this paper are included here and in the Appendix A. Additional data could be obtained from the authors upon request.

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
