# Peer review of "β-Lactam TRPM8 Antagonists Derived from Phe-Phenylalaninol Conjugates: Structure–Activity Relationships and Antiallodynic Activity"

_ijms, 2023, doi:10.3390/ijms241914894_

Round 1

Reviewer 1 Report

In the manuscript entitled, "β-Lactam TRPM8 antagonists derived from Phe-phenylalaninol conjugates: structure-activity relationships and antiallodynic activity", authors have designed and synthesized β-Lactam based Phe-phenylalaninol conjugates having inhibitory potential for TRPM8 receptor for antinociceptive property followed by the molecular mechanistic establishment and SAR by docking analysis. Overall, the research presents valuable insights into the development of TRPM8 antagonists and their potential role in alleviating allodynic symptoms. Here are some suggestions to improve the quality of the manuscript:

1.      Docking protocol needs to be validated with the appropriate ligand to enhance the authenticity of the computational prediction.

2.      The two-dimensional interactions will be more appropriate to develop SAR of the proposed compound which is missing in the manuscript.

3.      The quality of the figures should be improved for clarity of the obtained results.

4.      The comparative result analysis for the experimental outcome with their computational analysis should enhance the weightage of mechanistic hypothesis and proposed SAR.

Incorporating these review comments should help enhance the overall quality and impact of your research article. Good luck with your revisions and the submission process.

Minor editing of English language required

Author Response

In the manuscript entitled, "β-Lactam TRPM8 antagonists derived from Phe-phenylalaninol conjugates: structure-activity relationships and antiallodynic activity", authors have designed and synthesized β-Lactam based Phe-phenylalaninol conjugates having inhibitory potential for TRPM8 receptor for antinociceptive property followed by the molecular mechanistic establishment and SAR by docking analysis. Overall, the research presents valuable insights into the development of TRPM8 antagonists and their potential role in alleviating allodynic symptoms. Here are some suggestions to improve the quality of the manuscript:

 Thank you very much for the positive comments on this work

  1. Docking protocol needs to be validated with the appropriate ligand to enhance the authenticity of the computational prediction.

Thank you for this suggestion. We have now performed new docking experiments for menthol, the prototype agonist of TRPM8, and for AMTB, a model antagonist, whose complex structure with the channel is known (Cryo-EM). While for menthol docking studies suggested a good location into the agonists binding site and a closed pocket (subsite 6), only an 11% of solutions are located at the Cryo-EM antagonist’s pocked (subsite 6). Re-evaluation of our compounds at this S1-S4/TRP domain cavity revealed that β-lactam 35 have similar percentage of solutions as AMTB, while compounds 34 and 37 showed lower <5% solutions (see new Table 6). According to these figures, we have included new text in the corresponding subsection (L406-408 and L425-435 and L491-497).

  1. The two-dimensional interactions will be more appropriate to develop SAR of the proposed compound which is missing in the manuscript.

We did not consider a two-dimensional representation of the interactions due to the multiple binding sites suggested by the docking studies, and the different poses observed for compounds 34, 35 and 37. We considered that the tridimensional representation would give a better idea not only on the most important protein residues, but on empty space around the different substituents. SARs have been commented already in the corresponding sections on 1,4,4-trisubstituted and 1,3,4,4,tetrasubstituted b-lactams, near the tables. See also the response for point 4.

  1. The quality of the figures should be improved for clarity of the obtained results.

The figures have been modified for clarity

  1. The comparative result analysis for the experimental outcome with their computational analysis should enhance the weightage of mechanistic hypothesis and proposed SAR.

A comparative analysis interconnecting theoretical and experimental results has been included in the revised version (L498-513)

Reviewer 2 Report

The authors here tested the role of some β-Lactam analogs on TRPM8 channels to develop better inhibition efficacy on these channels. Drug synthesis, electrophysiological experiments, animal tests, and molecular modeling were used in the study. They found some of these TRPM8 antagonists are selective for TRPM8 channels. Four main binding sites of these compounds on the channel were revealed. In addition, a couple of compounds showed antinociceptive activity in a mouse model with cold allodynia. The manuscript is well-organized, and the methods are appropriate. There are some comments below.

1. Line 255, Why did the compounds show different inhibition on hTRAM8 and rTRAM8 channels? Is there a huge sequence difference between hTRAM8 and rTRAM8? What isoform structure was used in Figure 5?

2. Line 261, the reference to using menthol and how menthol activates TRPM8 channels should be mentioned or even discussed more in the text.

3. Where does menthol bind to the channel? Would menthol affect these compounds' effects on the channel?

3. Figure 2A, Why were there negative currents at resting/negative potentials?

4. Table 5, What does % binding at 10 uM mean? How is it different from % channel inhibition?

5. Please check some typos and grammar errors, like lines 299, 342, and 647. 

Please check for some typos and grammar errors. 

Author Response

The authors here tested the role of some β-Lactam analogs on TRPM8 channels to develop better inhibition efficacy on these channels. Drug synthesis, electrophysiological experiments, animal tests, and molecular modeling were used in the study. They found some of these TRPM8 antagonists are selective for TRPM8 channels. Four main binding sites of these compounds on the channel were revealed. In addition, a couple of compounds showed antinociceptive activity in a mouse model with cold allodynia. The manuscript is well-organized, and the methods are appropriate. There are some comments below.

  Thank you very much to the reviewer for the positive comments on this work

  1. Line 255, Why did the compounds show different inhibition on hTRAM8 and rTRAM8 channels? Is there a huge sequence difference between hTRAM8 and rTRAM8? What isoform structure was used in Figure 5?

hTRPM8 and rTRPM8 isoforms are highly homologous, as seen in Fig S1, which has been incorporated within the supplementary information for a better comprehension. No differences were found neither within the residues in the menthol/icilin binding sites nor in the five subsites suggested by our docking studies. Therefore, the different inhibition between isoforms could be related to different regulation (A clarifying sentence was included in the manuscript_R1, L262-266). This difference is not particular of our compounds, but also of other antagonists, as already indicated and referenced within the text (ref. 40).

rTRPM8 was used in Fig 5. Now it is indicated in the legend.

  1. Line 261, the reference to using menthol and how menthol activates TRPM8 channels should be mentioned or even discussed more in the text.

The binding pocket between menthol and TRPM8 involved residues of the S1 and S4 transmembrane helices (residues Tyr 745, R842, I846 and L843 have been found crucial for the activation of the channel). This has been commented within the text (L277-281, manuscript_R1). New reference 41, has been added.

  1. Where does menthol bind to the channel? Would menthol affect these compounds' effects on the channel?

For the binding pocket of menthol, see previous response. This binding pocket is away from the main binding sites of compounds 34, 35 and 37 defined by our docking studies.

Menthol was used as the agonist for activating the chanel. In the presence of the antagonist b-lactams, a concentration-dependent reduction of Ca2+ entry is observed, which was measured by a intramolecular calcium fluorescent probe.

  1. Figure 2A, Why were there negative currents at resting/negative potentials?

The presence of negative currents at negative potentials is due to the permeability of the TRPM8 ion channel to Na+ ions, which go into the cell based on the concentration gradient on the difference in concentrations on both sides of the membrane.

At positive potentials a clear increase in outward currents is observed due to the outflow of K+ ions, based on the difference in concentrations on both sides of the membrane.

The ionic concentrations in the patch clamp experiments are:

The intracellular pipette solution contained (in mM) 140KCl, 5 EGTA, and 10 mM HEPES, adjusted to pH 7.2 with KOH, and the extracellular solution contained (in mM) 140NaCl, 2CaCl2, 1MgCl2, 10D-glucose and 10 HEPES, adjusted to pH7.4 with NaOH. This paragraph has been included in the experimental procedures.

  1. Table 5, What does % binding at 10 uM mean? How is it different from % channel inhibition?

For TRP channels we performed functional assays, so what we measured is the percentage of inhibition of the channel activation induced by a stimulus (menthol in this case).

For the GPCR receptors CB2, CGRP and M3, we have measured the displacement by the compound (10uM) of specific radioligands, followed by the calculation of the corresponding inhibition percentage.

  1. Please check some typos and grammar errors, like lines 299, 342, and 647. 

Line300 implicated

Line 294 : 2 of

Line 226 trestments

Line 205 electronis

Lin 196 wasn

Thank you very much for the indicated grammar error and typos. All of them, and some others, have been corrected in the revised manuscript.

Reviewer 3 Report

The current manuscript by Cristina Martín-Escura et al. reported a serious of β–lactam derivatives with potent inhibiting for TRPM8 channel. The inhibitory effect of the compound on TRPM8 channels was confirmed through calcium flux and single-cell patch-clamp techniques, respectively. The compound's antinociceptive activity was validated in a mice model of cold allodynia induced by oxaliplatin. As I am not well-versed in chemical synthesis, I refrain from commenting on that aspect of the research. However, this paper presents a comprehensive and systematic pharmacological study elucidating the inhibitory effect of the compound on TRPM8 channel current. I have a few suggestions for this manuscript.

1. Please provide the specific concentration of the agonist utilized in the study, such as the concentration employed to induce TRPM8 channel activation through Menthol in calcium flow experiments. Additionally, kindly mention the number of experimental repetitions conducted as indicated in TABLE 1-2.

2. The IC50 of the compound on TRPM8-mediated calcium influx is listed in the table1-2. What is the maximum inhibitory rate of the compound on calcium flow? Please provide.

3. Please provide a detailed description of the methodology employed to calculate the inhibition rate of calcium influx for the compound, utilizing either the maximum fluorescence intensity or the AUC of the curve of fluorescence change. It is preferable to present a representative fluorescence graph over time in order to elucidate the calculation of the inhibition rate.

4. Similarly, in the context of patch-clamp experiments, it is imperative to present a temporal profile illustrating the dynamics of current alterations.

5. I do not recommend to include the docking studies here, unless the authors can verify the predicted binding sites  with site-specific mutation experiments. 

Author Response

The current manuscript by Cristina Martín-Escura et al. reported a serious of β–lactam derivatives with potent inhibiting for TRPM8 channel. The inhibitory effect of the compound on TRPM8 channels was confirmed through calcium flux and single-cell patch-clamp techniques, respectively. The compound's antinociceptive activity was validated in a mice model of cold allodynia induced by oxaliplatin. As I am not well-versed in chemical synthesis, I refrain from commenting on that aspect of the research. However, this paper presents a comprehensive and systematic pharmacological study elucidating the inhibitory effect of the compound on TRPM8 channel current. I have a few suggestions for this manuscript.

Thank you very much for comments and suggestions to improve the manuscript.

  1. Please provide the specific concentration of the agonist utilized in the study, such as the concentration employed to induce TRPM8 channel activation through Menthol in calcium flow experiments. Additionally, kindly mention the number of experimental repetitions conducted as indicated in TABLE 1-2.

Menthol concentration used to activate the TRP8 channels was 100 mM in all cases. This is now indicated in table’s description.

Three separate experiments in triplicate were used in Ca2+ microfluorometry and Patch-Clamp assays. Now indicated within the text (L167-168).

  1. The IC50 of the compound on TRPM8-mediated calcium influx is listed in the table1-2. What is the maximum inhibitory rate of the compound on calcium flow? Please provide.

Except for compoundss with no IC50 values, the maximum inhibition rate of all other β-lactam derivatives on calcium flux was 100% at 50 mM. This is now indicated in the manuscript.

  1. Please provide a detailed description of the methodology employed to calculate the inhibition rate of calcium influx for the compound, utilizing either the maximum fluorescence intensity or the AUC of the curve of fluorescence change. It is preferable to present a representative fluorescence graph over time in order to elucidate the calculation of the inhibition rate.

In previous papers from the same Editorial, detailed experimental biological methods were considered plagiarism. For this reason, in this case, we opted by referencing previous papers in which these methods are detailed. Percentages of inhibition were calculated from intramolecular fluorescent intensities.

  1. Similarly, in the context of patch-clamp experiments, it is imperative to present a temporal profile illustrating the dynamics of current alterations.

Thank you for noting this. In the revised manuscript we have included the time course of activation by menthol and the inhibition by compound 34 at two concentrations (Figure 2C).

  1. I do not recommend to include the docking studies here, unless the authors can verify the predicted binding sites with site-specific mutation experiments. 

As we identify four-five main possible subsites for our compounds, the number of mutants to be prepared are huge, and cannot be done in the participant’s facilities. Now we have included AMTB and menthol in docking studies, as controls, and commented the results according to the obtained figures (L425-435 and L491-497). 

Round 2

Reviewer 2 Report

The authors have addressed all my concerns.